

# Seed type, habitat and time of day influence post-dispersal seed removal in temperate ecosystems

Katja Wehner,  Lea Schäfer,  Nico Blüthgen and  Karsten Mody

Ecological Networks, Technical University of Darmstadt, Darmstadt, Germany

## ABSTRACT

Seed survival is of great importance for the performance of plant species and it is strongly affected by post-dispersal seed removal by either different animals such as granivorous species and secondary dispersers or abiotic conditions such as wind or water. The success of post-dispersal seed removal depends on seed specific traits including seed size, the presence of coats or elaiosomes, the mode of seed dispersion, and on the habitat in which seeds happen to arrive. In the present study we asked how seed traits (dehulled vs. intact; size; dispersal mode), habitat (forest vs. grassland), and time of day (night vs. day) influence post-dispersal seed removal of the four plant species *Chelidonium majus, Lotus corniculatus, Tragopogon pratensis* and *Helianthus annuus*. Seed removal experiments were performed in three regions in Hesse, Germany. The results showed different, inconsistent influences of time of day, depending on habitat and region, but consistent variation across seed types. *C. majus* and dehulled *H. annuus* seeds had the fastest removal rates. The impact of the habitat on post-dispersal seed removal was very low, only intact *H. annuus* seeds were removed at significantly higher rates in grasslands than in forests. Our study demonstrates consistent differences across seed types across different habitats and time: smaller seeds and those dispersed by animals had a faster removal rate. It further highlights that experimental studies need to consider seeds in their natural form to be most realistic.

## INTRODUCTION

Seed dispersal, survival and germination are crucial for plant reproduction. Whereas seed dispersal depends on seed size, dispersal mode, and annual seed production (*Lambert & Champman, 2005*), seed survival is additionally affected by post-dispersal seed removal. In general, the term *post-dispersal seed removal* includes both seed predation and secondary seed dispersal (*Vander Wall, Kuhn & Beck, 2005*). For most plant species, the relative rate of predation (seeds that are digested by granivorous animals) versus dispersal (seeds that are consumed or removed but survive and germinate) is unknown (*Vander Wall, Kuhn & Beck, 2005*). Seed predation can limit the population growth of certain plant species (*Menalled et al., 2000*), and it varies considerably due to different factors (*Hulme, 1994*), including habitat (*Notman, Gorchov & Cornejo, 1996*; *Holl & Lulow, 1997*), microhabitat (*Manson & Stiles, 1998*), seed species (*Borchert & Jain, 1978*), seed burial (*Hulme & Borelli,*

Corresponding author
Katja Wehner, kdwehner@gmx.de

*1999*) and seed density (*Myster & Pickett, 1993*). By feeding on seeds, granivores influence plant species diversity (*Schupp, 1988*), plant community structure (*Brown & Heske, 1990*) and patterns of plant recruitment (*Borchert & Jain, 1978*).

Besides post-dispersal seed predation, secondary seed dispersal also determines the performance of a plant, e.g., by reducing seed and seedling mortality near the parent, or by carrying seeds to suitable microhabitats for establishment and growth (*Howe & Smallwood, 1982*; *Meyer et al., 2017*). Although seed predation is a common fate of seeds encountered by granivores, a number of animal groups transport seeds to microsites that favor seedling establishment; often up to half of the removed seeds are dispersed and germinate (*Vander Wall, Kuhn & Beck, 2005*). In this case, seed removal is considered to be a mutualistic interaction between plants and animals (*Bond, 1994*) because seed dispersal benefits plants by reducing density-dependent seed and seedling mortality (*Harms et al., 2000*). Furthermore, seed dispersal reduces competition with the parent plant and allows the exploitation of suitable new habitats (*Meyer et al., 2017*), which is an important process in the reproductive cycle (*Vander Wall & Longland, 2004*).

Many animal groups such as rodents (*Forget & Milleron, 1991*), birds (*Levey et al., 2005*), ants (*Bond, 1983*; *Peters, Oberrath & Böhning-Gaese, 2003*), dung beetles (*Andresen, 2002*) and carabids (*Brust & House, 1988*), as well as wind (*Tackenberg, Poschlod & Bonn, 2003*) and water (*Kowarik & Säumel, 2008*) can move seeds after primary dispersal from the mother plant. Secondary dispersal by ants is well known and some plants produce seeds with lipid rich appendages (elaiosomes) that are highly attractive to ants and facilitate dispersal (*Handel & Beattie, 1990*).

In many removal studies, easily obtainable dehulled sunflower seeds are used (e.g., *Meyer et al., 2017*). As dehulled seeds are rarely found in nature, this study attempts to clarify the effect of the presence of seed coats on seed removal. One of the main functions of the seed coat is the protection of the embryo against mechanical injuries (*Souza & Marcos-Filho, 2001*). Furthermore, seed size affects seed dispersal and has strong effects on the range of potential granivores (*Xiao, Zhang & Wang, 2005*). In rodent-dispersed fagaceous species, for example, the distribution range of seeds increased with seed size, and larger seeds were more often recaptured after consumption than smaller seeds (*Xiao, Zhang & Wang, 2005*).

The occurrence and activity of seed consuming animals may vary between different habitats (*Webb & Willson, 1985*; *Lindgren, Lindborg & Cousins, 2018*) and times of day (*Miranda-Jácome & Flores, 2018*). Studies on post-dispersal seed removal in different habitats revealed significant differences in the proportion of total seeds remaining in open pastures, forests and beneath isolated pasture trees, and different seed species suffered differently in different habitats (*Holl & Lulow, 1997*). Seed-feeding animals such as ants, rodents and birds have a great potential to influence seed dynamics (*Hulme, 1994*). Since their occurrence and abundances differ among habitats, their influence on seed removal may differ accordingly.

The time of day may also influence post-dispersal seed predation by temporal differences in foraging activities of seed predators. These differences may develop to reduce competition between different seed predators (*Brown et al., 1975*), to reduce enemy pressure (*Manson & Stiles, 1998*), or to take advantage of favorable abiotic conditions (*Whitford et al., 1981*).

In the present study we asked how seed traits (dehulled vs. intact; size; dispersal mode), habitat (forest vs. grassland), and time of day (night vs. day) influence post-dispersal seed removal. We did not distinguish between predation and dispersal since seed removal has not been tracked, but seed predators/dispersers have been incidentally observed. As seed predators and seed dispersers differ in their preferences for seed size (*Brown et al., 1975*; *Reader, 1993*; *Larios et al., 2017*) and in spatial and temporal foraging activities, we expected that seed traits, habitat, and time of day contribute to variation in seed removal rates.

To examine the effects of seed traits, habitat, and time of day on post dispersal seed removal, we used the three native plant species *Chelidonium majus* L., *Lotus corniculatus* L. and *Tragopogon pratensis* L., and the crop and ornamental plant *Helianthus annuus* L. These plant species were chosen to represent a large variation in seed size and dispersal mode. For *H. annuus*, we used intact fruits (intact cypselae consisting of kernel surrounded by seed coat), and in addition dehulled kernels. *H. annuus* seeds are generally dispersed by animals which usually use them as food and can also be blown to different localities by wind (*Cummings & Alexander, 2002*). The seeds of *T. pratensis* are generally wind-dispersed with a feathery pappus as flying organ (*Casseau et al., 2015*). *C. majus* seeds are small and associated with lipid rich appendages (elaiosomes) that are highly attractive to ants (*Handel & Beattie, 1990*). Seeds of *L. corniculatus* are small and catapulted up to five meters by a longitudinally dehiscent fruit (*Jones & Turkington, 1986*).

We expected post-dispersal seed removal to differ with seed traits: smaller seeds (*C. majus*, *L. corniculatus*) should be removed faster than larger seeds (*H. annuus*), and we expected a lower removal of intact compared to dehulled sunflower (*H. annuus*) seeds due to the protective properties of the seed coat. Concerning the mode of seed-dispersal, we expected that seeds that are typically wind-dispersed (*T. pratensis*) are less affected by secondary seed dispersal via animals than seeds with elaiosomes (*C. majus*), which are generally considered being ant-dispersed (*Handel & Beattie, 1990*; *Fonara & Dalling, 2005*). We further expected a higher seed removal rate in grasslands as compared to forests due to higher abundances of granivores. In both habitats we expected that seed removal at different times of the day would be different depending on the seed type: large seeds consumed by nocturnal rodents (*H. annuus*) should be removed faster at night, while small seeds consumed by diurnal ants (*C. majus*) should be removed faster during the day.

## MATERIALS & METHODS

### Study area

Our study was conducted in the South of Hesse, Germany, in June and July 2018. Since we were interested in comparing the two common habitats forest and grasslands, we selected forest (coniferous and deciduous) and grassland sites in each of three regions which are 20 km apart on average: Darmstadt, Oberzent/Airlenbach and Bad König/Zell (see Supplemental Information 1 for site coordinates). In Darmstadt, all grassland sites are meadows that are mown once or twice per year. Forest sites are recreational areas comprising only a moderate amount of forestry; they are mixed forests with native deciduous trees. In Airlenbach, grasslands are intensively used: they are mown three times a year and fertilized

with cattle manure. Forest sites are recreational areas comprising mixed and coniferous stands. In Zell, grassland use is not agricultural but sites are mown once a year to avoid bush encroachment. Sites are located close to the river Mümling and regularly flooded. Forest sites are recreational deciduous mixed forests.

## Plant species and seed parameters

Four plant species with different dispersal modes and seed traits were used:

  a) *Chelidonium majus*; small seeds with elaiosomes, dispersed by arthropods

  b) *Lotus corniculatus*; small seeds lacking elaiosomes, dispersed by longitudinally dehiscent fruits

  c) *Tragopogon pratensis*; medium sized seeds with a pappus, wind dispersed

  d) *Helianthus annuus*; large seeds dispersed by animals (birds, arthropods) and wind, intact (with seed coat) and dehulled seeds were compared.

Accordingly, five seed types were tested in total, represented by a single seed species or in case of *H. annuus* two variants of seed species. All seeds were purchased at Rieger-Hofmann GmbH, Blaufelden-Raboldshausen, Germany. For each seed type, the dry mass was quantified by measuring the mass of ten randomly selected seeds and the mean and standard deviation for each seed type was calculated: *C. majus* $= 0.7 \pm 0.0$ mg, *L. corniculatus* $= 1.1 \pm 0.1$ mg, *T. pratensis* $= 7.9 \pm 2.5$ mg, *H. annuus* intact $= 83.1 \pm 18.9$ mg, *H. annuus* dehulled $= 50.4 \pm 7.0$ mg.

### Study 1: Seed removal, habitat and seed type

We compared the seed removal of different seed types in two forest and two meadow sites in Darmstadt, and one meadow and one forest site in Airlenbach. For each site, five subplots were established 10 m away from each other. We checked plates for the number of remaining seeds at regular intervals of 60 min for ten hours in Airlenbach and for seven hours in Darmstadt to investigate the removal rate and after 48 h for total removal. We counted a seed as removed when it had left the plate; thus, we recorded a seed as present, when it had been removed from its well, but was still lying on the plate (*Meyer et al., 2017*). Based on the number of seeds remaining after different time intervals, we calculated the percentage of removed seeds. While counting seeds at different time intervals, incidental observations on seed predators/dispersers (e.g., ants, slugs, other arthropods) were recorded.

A total of 25 seeds of the same seed type were offered simultaneously within a subplot on a 12 cm $\times$ 12 cm $\times$ 0.5 cm (length $\times$ width $\times$ height) gray plastic plate. The plate contains 25 evenly spaced wells for seed exposure, each well containing a single seed. For each seed type a separate plate was used; this method is consistent with standard protocols for ecosystem process assessments (*Meyer et al., 2017*). Seed plates were haphazardly placed flat on the ground approximately in the center of the site by carefully flattening the groundcover.

### Study 2: Seed removal and time of day

We investigated the influence of time of day (day vs. night) on seed removal in three different regions (Darmstadt, Airlenbach and Zell). In total, we tested 16 different study

sites per habitat: six grasslands and six forests in Darmstadt, six grasslands and six forests in Airlenbach and four grasslands and four forests in Zell. In each study site, we tested four different seed types: *C. majus, L. corniculatus,* intact *H. annuus* and dehulled *H. annuus* seeds. As in study 1, 25 seeds of the same seed type were offered simultaneously on a grey plastic plate. Seeds were left in the field for 12 h during the day. After 12 h, the remaining seeds were quantified. Afterwards, the plates were refilled with new seeds. To avoid positional effects, the plates were set up 10 m away from the day plot; all plate locations were chosen haphazardly. The seed plates were left in the field for another 12 h, covering a period of reduced light and the night. As the 12-hours interval did not perfectly match the actual day- and night-period, observations during the ''daytime'' assessments always reflected daytime conditions, but the observations during the ''nighttime'' assessments also included evening and morning (twilight) conditions. This means that species that depend on light and higher temperatures were only recorded during the daytime assessments, but the species recorded during the nighttime assessments may also include species that are active at the transition between night and day. After 12 h, remaining seeds were quantified. Removal rates were calculated as percentages.

## Statistical analysis

### Study 1: Seed removal, habitat and seed type

All collected data were analyzed with R 3.5.2 (*R Core Team, 2018*). Data were checked for normal distribution and log(x+1) or square root transformed to increase homogeneity of variances if necessary.

To test the effect of seed type on seed removal over time in different habitats (forest vs. grassland), we first compared the proportional removal of seeds after 48 h (*Meyer et al., 2017*). Removal rate was defined as $R = \log((N_{\text{removed}}/25)+1)$, with $N_{\text{removed}}$ representing the number of seeds removed after 48 h. Results were analyzed using a linear mixed-effect model (lmer) with $R$ as dependent variable, and region, habitat and seed type as independent variables, including the interaction between habitat and seed type. Plots were included as random factor, individual effects were analyzed by one-way ANOVA included in the car package (*Fox & Weisberg, 2019*).

We additionally investigated seed removal over time using linear regressions; first, we compared the fit of linear versus non-linear models using Integrated Nested Laplace Approximation (Inla) and results showed linear models to fit best. For each seed type, seed removal (dependent variable) was plotted versus time (predictor). Effects were analyzed using a linear mixed-effect model (lmer) with square root transformed number of seeds removed as dependent variable and region, habitat, time and seed-type as independent variables, including interaction terms between variables. Plots were included as random factor (see Supplemental Information 2 for raw data).

### Study 2: Seed removal and time of day

To evaluate the influence of time of day (day vs. night), we also used a linear mixed-effect model (lmer) with time of day, habitat and region as independent variables (including the interaction terms of the variables) and plots as random factor. One-way ANOVA

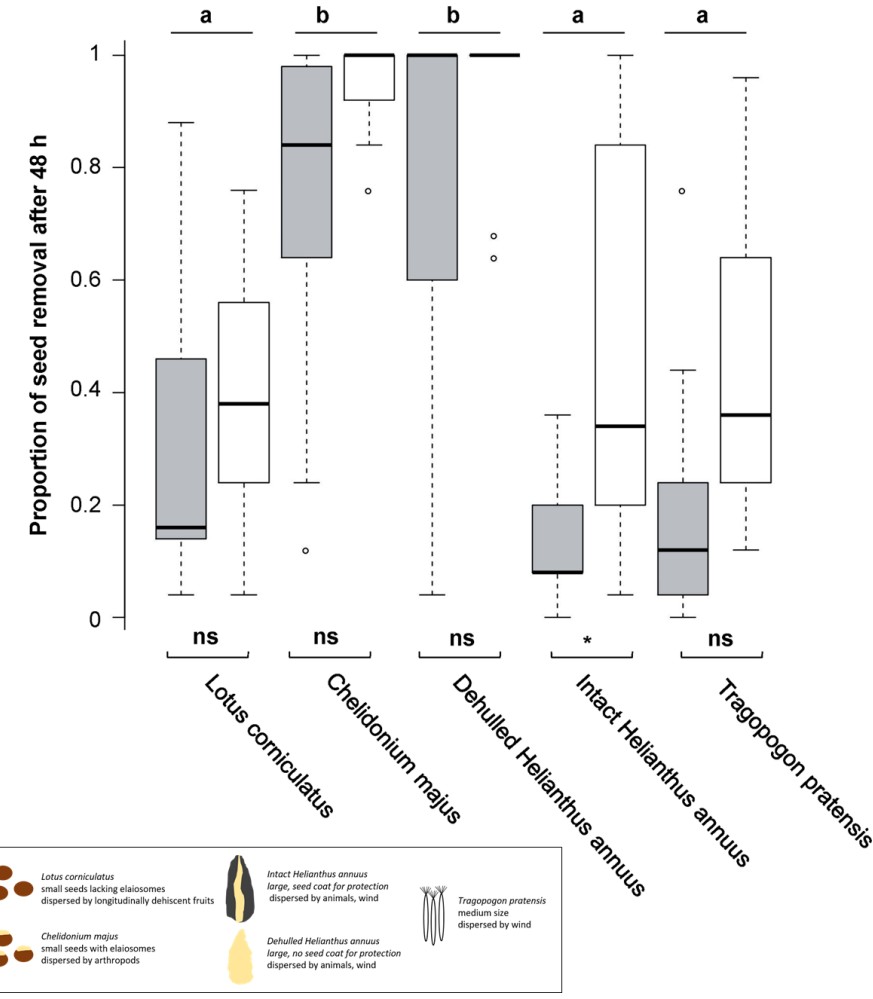

**Figure 1** **Seed removal after 48 h.** Proportional seed removal of *Lotus corniculatus*, *Chelidonium majus*, dehulled and intact *Helianthus annuus* and *Tragopogon pratensis* in forests (grey) and grasslands (white) after 48 h. Different letters above boxplots indicate significant differences between seed types, significance levels beneath boxplots indicate differences between habitats, whiskers denote range of data. The black box summarizes differences in seed traits. ns, not significant; **, $p < 0.01$, (*), $p = 0.05$.

was calculated individually for each seed type per habitat and region (see Supplemental Information 3 for raw data).

## RESULTS

### Study 1: Seed removal, habitat and seed type

Seed removal after 48 h differed significantly across seed types but not among regions and only slightly among habitats; however, there was no interaction between habitat and seed type (Fig. 1, Table 1). Both in forests and in grasslands, *C. majus* and dehulled *H. annuus* seeds were often removed entirely, and their removal was significantly higher than for *L. corniculatus,* intact *H. annuus* and *T. pratensis* seeds; the latter three species did not differ significantly (Fig. 1). Removal of *L. corniculatus*, *C. majus*, dehulled *H. annuus* and

**Table 1   Post-dispersal seed removal over time.** Influence of seed type, habitat and region on post-dispersal seed removal after 48 h, tested by a linear mixed-effects model on log+1 transformed data with subplots as random factor.

|  | Df | Chisq | *p* |
|---|---|---|---|
| seed type | 4 | 152.247 | **<0.001** |
| habitat | 1 | 3.374 | 0.066 |
| region | 1 | 0.300 | 0.584 |
| habitat:seed type | 4 | 5.201 | 0.267 |

*T. pratensis* seeds did not differ significantly between habitats, whereas seeds of intact *H. annuus* were removed faster in grasslands than in forests.

For each seed type in each region, seed removal increased continuously over time (Fig. 2, Table 2). In both regions and both habitats, *C. majus* and dehulled *H. annuus* seed removal was fastest, corresponding to the results after 48 h. With one exception (forest, Darmstadt), *C. majus* seed removal had the highest increase over time, while the removal of dehulled *H. annuus* seeds was fastest in forests in Darmstadt. However, removal rates changed in time with habitat and seed type (Table 2); e.g., *T. pratensis* was removed at higher rates in grasslands in Darmstadt but in forests in Airlenbach.

## Study 2: Seed removal and time of day

Post-dispersal seed removal at different times of day significantly differed with habitat and region (Table 3). The removal of *C. majus* was higher at night in grasslands in Darmstadt and forests in Zell, but more pronounced during the day in grasslands in Airlenbach and Zell (Fig. 3B). Night removal of dehulled *H. annuus* was higher in grasslands in Darmstadt (Fig. 3C) and night removal of intact *H. annuus* was more pronounced in forests in Zell (Fig. 3D).

In general, seed removal of *L. corniculatus* was generally low (Fig. 3A). *C. majus* was removed at higher rate at night in grasslands in Darmstadt and in forests in Zell, whereas day-removal seemed to be more pronounced in forests in Airlenbach and grasslands in Zell (Fig. 3B). The removal of dehulled *H. annuus* was generally low in Darmstadt, but higher in grasslands in Airlenbach and forests in Zell (Fig. 3C). Intact *H. annuus* was even less removed than dehulled seeds, but tended to be removed at higher rate at night (Fig. 3D).

## DISCUSSION

### Study 1: Seed removal, habitat and seed type

Post-dispersal seed removal after 48 h differed across plant species. In both habitats, forest and grassland, seed removal of *C. majus* and dehulled *H. annuus* seeds was significantly higher than the removal of *L. corniculatus,* intact *H. annuus* and *T. pratensis* seeds.

In contrast to *L. corniculatus,* *C. majus* seeds are myrmecochorous because of the presence of elaiosomes (*Pemberton & Irving, 1990*; *Peters, Oberrath & Böhning-Gaese, 2003*). These lipid-rich appendages are attractive to many ant species and promote burial and dispersal (*Hughes, Westoby & Jurado, 1994*; *Fischer et al., 2008*). Since the abandoned seeds maintain their ability to germinate, seeds are dispersed effectively (*Kjellsson, 1985*;

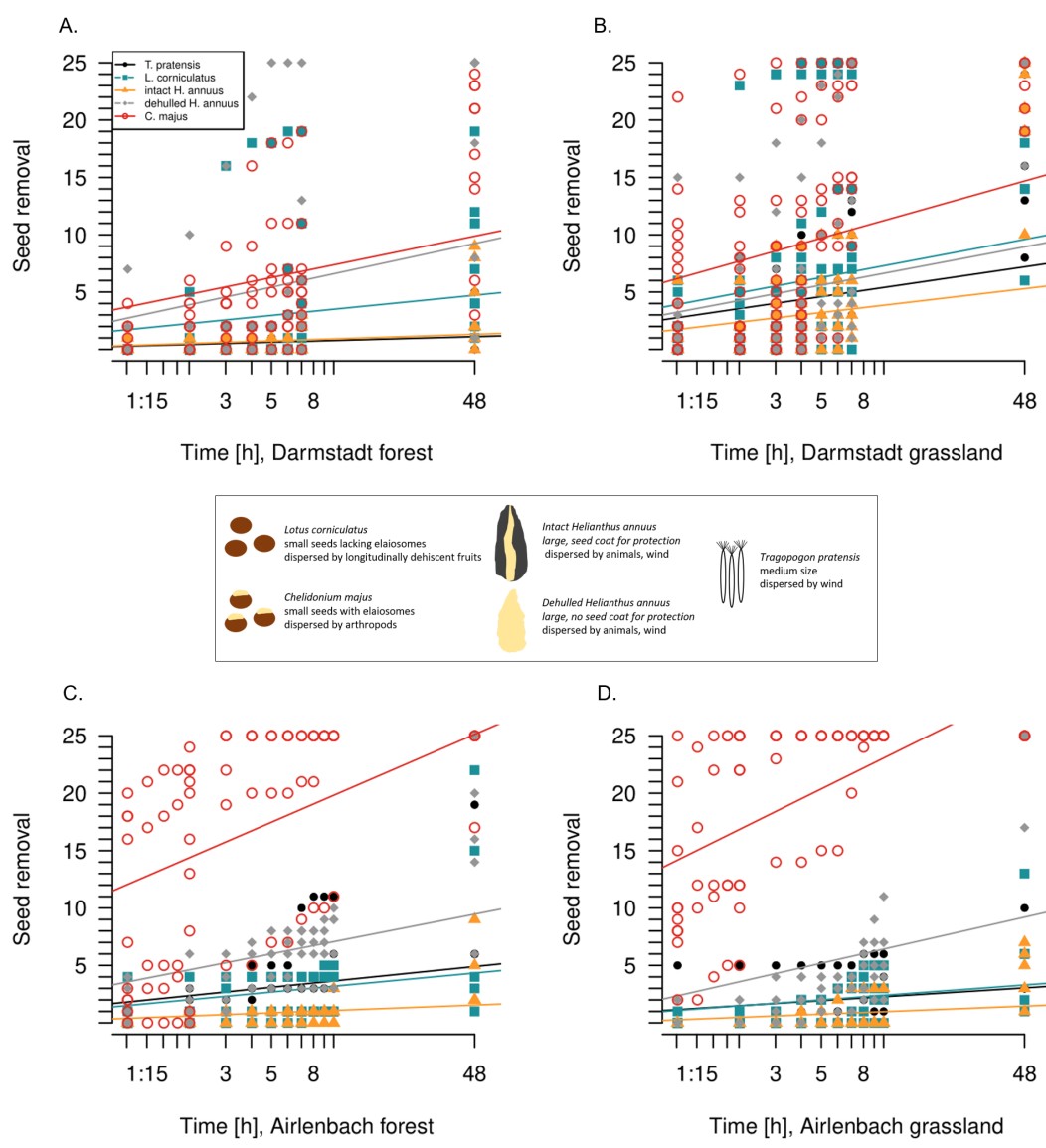

**Figure 2  Post-dispersal seed removal over time.** The 48 hours-timescale of seed removal of *Lotus corniculatus*, *Chelidonium majus*, dehulled and intact *Helianthus annuus* and *Tragopogon pratensis* in forests (A) and grasslands (B) in Darmstadt, and forests (C) and grasslands (D) in Airlenbach. Note the logarithmic scale of the *x*-axis. The black box summarizes differences in seed traits.

*Gorb & Gorb, 2003*). We observed that ants removed seeds of both *C. majus* and *L. corniculatus*, but preferred seeds of *C. majus*.

Removal of dehulled *H. annuus* was significantly higher than of natural, intact *H. annuus* seeds. This result mirrors the physical protection of the nutritional seed components by the seed coat (*Souza & Marcos-Filho, 2001*) that are apparent in dehulled seeds, and may additionally be driven by increased energy requirements for handling such intact seeds (*Pyke, Pulliam & Charnov, 1977*). In our study, we observed that ants and slugs consumed dehulled but ignored intact seeds, which was most probably directly related to

**Table 2  Post-dispersal seed removal over time.** Influence of region, habitat, time and seed type on post-dispersal seed removal over time tested by a linear mixed-effects model on sqrt-transformed data with subplots as random factor.

|  | DF | Chisq | p |
|---|---|---|---|
| region | 1 | 3.923 | **0.047** |
| habitat | 1 | 2.797 | 0.094 |
| time | 1 | 550.026 | **<0.001** |
| seed type | 4 | 785.526 | **<0.001** |
| habitat*time | 1 | 3.959 | **0.047** |
| habitat*seed type | 4 | 10.282 | **0.036** |
| time*seed type | 4 | 57.48 | **<0.001** |
| habitat*time*seed type | 4 | 3.551 | 0.470 |

**Table 3  Post-dispersal seed removal and time of day.** Influence of time of day, habitat and region on post-dispersal seed removal after 12 h, tested by a linear mixed-effects model on sqrt-transformed data with subplots as random factor.

|  | DF | Chisq | p |
|---|---|---|---|
| day time | 1 | 0.806 | 0.369 |
| habitat | 1 | 0.732 | 0.392 |
| region | 2 | 5.382 | 0.068 |
| day time*habitat | 1 | 0.002 | 0.969 |
| day time*region | 2 | 3.476 | 0.176 |
| habitat*region | 2 | 4.963 | 0.083 |
| day time*habitat*region | 2 | 12.451 | **0.002** |

the accessibility of the seed material in dehulled seeds. In addition to physical protection, dehulled seeds may produce more attractive odors, resulting in higher seed predation by granivores guided by olfactory cues (*Jaganathan, 2018*; *Vander Wall, 1998*).

*Tragopogon pratensis* seeds are attached to a pappus and are primarily wind-dispersed (*Casseau et al., 2015*). Thus, we expected the removal by animals to be of minor importance. As predicted, removal of *T. pratensis* seeds was low and no *T. pratensis* seed removal by animals was observed (but predators/dispersers were not monitored over time). However, secondary seed dispersal by animals may be highly relevant also for primarily wind-dispersed plant species (*Der Weduwen & Ruxton, 2019*), e.g., when wind-dispersed pines (*Pinus* spp.) are dispersed by scatterhoarding rodents (*Vander Wall, 2003*).

In summary, the results of our study show that post-dispersal seed removal depends on seed specific traits that differ among the seed species we used. Smaller seeds (*C. majus*, *L. corniculatus*), seeds that are more easily accessible (dehulled *H. annus*) and those dispersed by animals (*C. majus*) are removed faster than large (intact *H. annuus*), protected (intact *H. annuus*) or those seeds that depend on abiotic conditions for dispersal (*T. pratensis*). Since *C. majus* were removed at the highest rate, the presence of elaiosomes may have strengthened the advantage of being small. Thus, *C. majus* may have achieved an increased germination probability in new habitats away from the mother plant. However, the

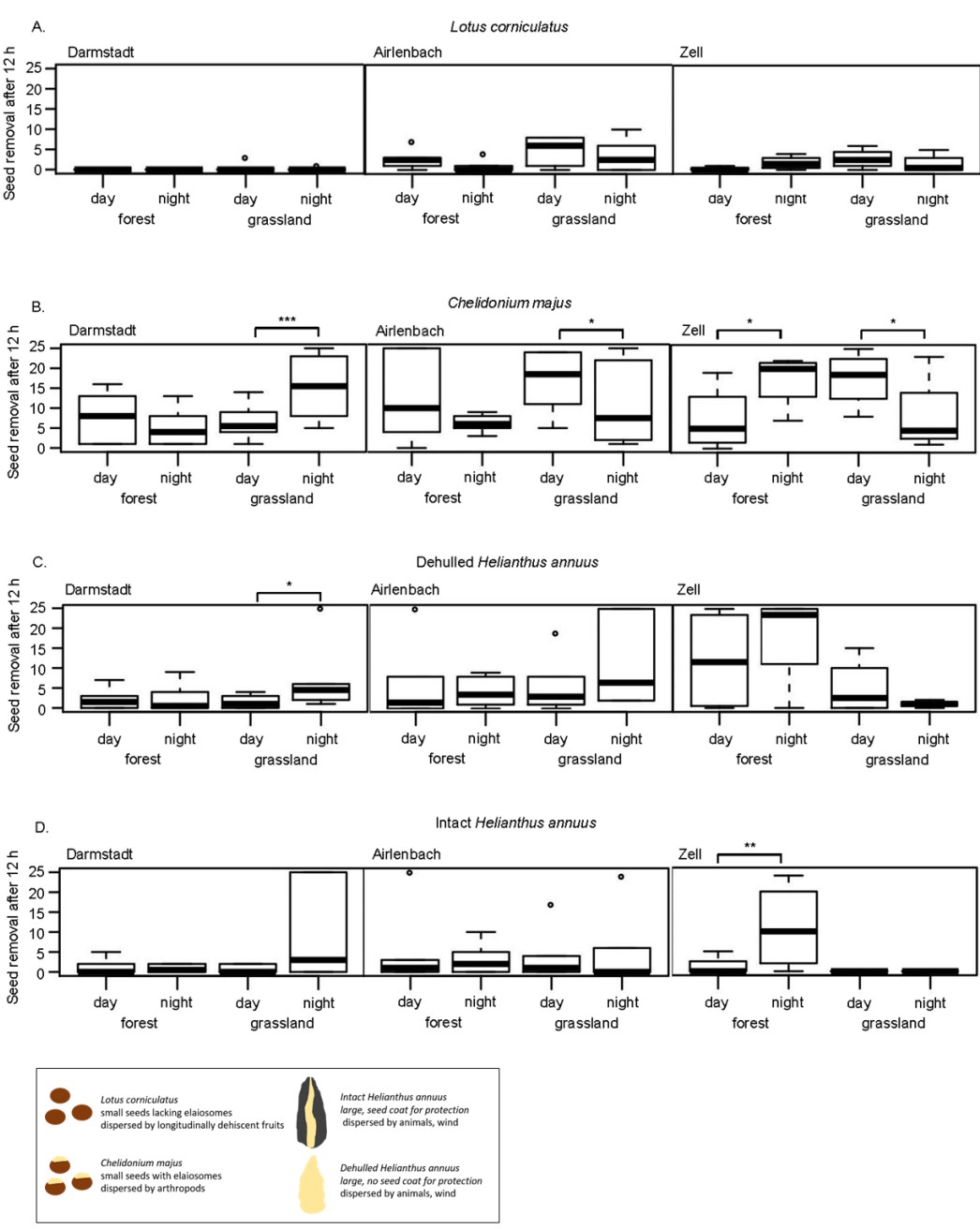

**Figure 3 Post-dispersal seed removal and time of day.** Removal of seeds of *Lotus corniculatus* (A), *Chelidonium majus* (B), dehulled (C) and intact (D) *Helianthus annuus* at different times of day (day vs. night) in forest and grassland sites in Darmstadt, Airlenbach and Zell. Whiskers denote range of data, the black box summarizes differences in seed traits. *, $p < 0.05$, **, $p < 0.02$, ***, $p < 0.01$.

variation of seed traits used in this study may not be exclusively causal for the observed results owing to the potential collinearity with other unmeasured traits.

Typically, habitat (forest vs. grassland) had no effect on seed removal, although removal of intact *H. annuus* was higher in grasslands. In general, as seed predators may differ in

spatial foraging patterns (*Brown et al., 1975*), many studies indicate a relevant effect of habitat type on seed removal concerning both microhabitats (*Notman, Gorchov & Cornejo, 1996*; *Ji-Qi & Zhi-Bin, 2004*) and forest and grassland sites (*Holl & Lulow, 1997*). Previous studies comparing levels of seed removal in early- and late-successional habitats have shown variable results: a few studies indicated that seed predation is higher in early than in mature successional habitats (*Uhl, 1987*; *Hammond, 1995*) whereas other studies reported opposite results (*Aide & Cavelier, 1994*; *Osunkoya, 1994*). Other studies showed that the habitat with the highest level of seed predation varies with seed species (*Willson & Whelan, 1990*; *Whelan et al., 1991*; *Holl & Lulow, 1997*). However, the effect of habitat on post-dispersal seed removal seems inconsistent and may depend on seed type and predator. Therefore, a comprehensive seed study of the plant community in different habitats and seed predators therein may be necessary to solve that problem.

**Study 2: Seed removal and time of day**
The effects of time of day on post dispersal seed removal of seed types differed with habitat and region. Contrary to this finding, several former studies indicated that seed predators differ not only in their preferences for seed types but also in their temporal patterns of foraging activities (*Brown et al., 1975*). Granivorous rodents are mainly nocturnal (*Abramsky, 1983*; *Miller, 1994*; *Xiao, Zhang & Wang, 2005*), whereas harvester ants are mostly diurnal (*Abramsky, 1983*; *Díaz, 1992*), as ectothermy of ants renders foraging activities temperature-dependent (*Whitford et al., 1981*; *MacKay & MacKay, 1989*). Thus, we expected that time of day influences seed removal, and that removal of myrmecochorous *C. majus* seeds was higher at day which was true for grassland sites in Airlenbach and Zell. Contrary to this expectation, removal of *C. majus* seeds was even higher at night in grasslands in Darmstadt and forests in Zell. Possibly, unusually very high temperatures in the summer 2018, may have shifted foraging activities of diurnal granivores to the early morning and late evening hours, which were covered by the nighttime assessments.

# CONCLUSION

Our study demonstrates that seed type and—to a lesser extent—habitat influence seed removal. With regard to studies on seed removal, our studies emphasize that the traits of the used seeds, like in our case the comparison of intact and dehulled seeds, may have strong effects on the outcome of experiments. Furthermore, differences between day and night removal should be considered.

# ACKNOWLEDGEMENTS

We thank E. Schäfer for providing the trial areas in Airlenbach and N. Simons for supporting the statistical analyses. We are further grateful to two anonymous reviewers for very valuable comments on an earlier draft of the manuscript.

### Funding

The study was supported by the German Research Foundation and the Open Access Publishing Fund of Technische Universität Darmstadt. The funders had no role in study design, data collection and analysis, decision to publish, or preparation of the manuscript.

### Grant Disclosures

The following grant information was disclosed by the authors:
German Research Foundation.
Open Access Publishing Fund of Technische Universität Darmstadt.

### Competing Interests

The authors declare there are no competing interests.

### Author Contributions

- Katja Wehner conceived and designed the experiments, analyzed the data, prepared figures and/or tables, authored or reviewed drafts of the paper, and approved the final draft.
- Lea Schäfer performed the experiments, analyzed the data, prepared figures and/or tables, authored or reviewed drafts of the paper, and approved the final draft.
- Nico Blüthgen conceived and designed the experiments, analyzed the data, prepared figures and/or tables, and approved the final draft.
- Karsten Mody conceived and designed the experiments, analyzed the data, authored or reviewed drafts of the paper, and approved the final draft.

### Data Availability

   The raw measurements are available in the Supplementary Files.

### Supplemental Information

Supplemental information for this article can be found online at http://dx.doi.org/10.7717/peerj.8769#supplemental-information.

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
