# Peer review of "Seed type, habitat and time of day influence post-dispersal seed removal in temperate ecosystems"

_PeerJ, doi:10.7717/peerj.8769_

## Round 0.1 · original submission · Major Revisions

The reviewers and I appreciate the effort that when into this manuscript. However, there are several sections that could use further clarification, or be better placed within the context of this study. The reviewers did a great job of identifying these places that could use clarification, as well as where the messaging could be honed to best fit the study. Please thoroughly address their comments.

Reviewer 1 ·

Basic reporting

This is a well-research manuscript which makes good use of the wide and varied literature regarding seed removal. The introduction and background give a good explanation of where the research currently stands, and why this study is important. However, there are sections where the language is unclear, or where more explanation would aid the reader. Possibly contrasting the findings of this paper with those reported in Fornara & Dalling, 2005, Journal of tropical Ecology, may be interesting, but are not essential. The data is supplied, and the figures are clear for the most part, although some of the symbols could be explained further and made larger to ease reading. Please find below more detailed information on suggested changes:

Line 38, 326 – Lambert & Champmann/Champman, 2005 – please check the spelling for this reference
Line 42 – “relative rate” instead of “relative importance”
Line 44 – “varies considerably” instead of “considerably varies”
Line 71 – missing the words “the presence of” in the section “the effect of ___ seed coat”
Line 73-74 – not clear how seed size plays a role when this paragraph has discussed the effect of the hull – perhaps clarify the relationship between dehulling and seed size more clearly.
Line 84 – section in parentheses can be its own sentence, but please provide justification e.g. difficulty of tracking seeds etc.
Lines 90-96 – please detail the mode of dispersal and basic morphology (especially any adaptations for modes of dispersal) for all species
Line 99 – why do you expect a lower rate of removal for intact sunflower seeds?
Line 102 – another good source to back up this claim is Fornara & Dalling, 2005, Journal of Tropical Ecology
Line 190 – “during the day” instead of “at day”
Line 194 – “removed at a higher rate” instead of “strongly removed”
Line 196 – “higher” instead of “stronger”
Line 198 – “removed at higher rate” instead of “stronger removed”
Lines 212-214 – this is a very good explanation. Perhaps refer to these points at line 99 to explain why you expect a lower rate of removal for hulled sunflower seeds.
Line 220 – “primarily” instead of “primary”
Line 230 – “removed at the highest rate” instead of “removed best”
Line 231 – perhaps clarify how seed removal is an advantage, e.g. it increases the chances of dispersal away from the parent.
Line 242 – “inconsistent” instead of “inconstant”, and “may depend on” instead of “depends on”
Line 373 – author’s initials are punctuated, unlike rest of reference list – please check for consistency
Line 375 – font appears to change half-way through reference – please check for consistency

Figure 1 – It is not entirely clear what the ns and * below the bars mean. Please provide a more detailed explanation in the figure legend. I assume it indicates whether the differences in seed removal between habitats were significant, but I have no way of confirming this solely by looking at the graph and the figure legend.
Figure 2 – I am unable to distinguish between the lines on the graph. Perhaps use different types of dotted or dashed lines instead of shades of grey. Also, the legend is very hard to read and it is not obvious that the white-filled circles are C. majus. Perhaps increase the legend size.
Figure 3 – please indicate what the asterisks mean in the figure legend and increase their font size for ease of reading.

Experimental design

This manuscript reports the results of a well-thought-out experiment. The research question is clear and meaningful, and the rationale is well-thought-out. The methods are very well-described, although some further information and justification may be useful to the reader. Please find the details of suggested changes below:

Lines 118-120 – please provide the modes of dispersal for Tragopogon pratensis and Helianthus annuus
Line 142 – please consistently use either “meadow” or “grassland”, unless there is a difference between the two. If there is a difference, please explain what it is (although line 112 appears to indicate that they are the same)
Lines 142-143 – why was T. pratensis not used for this experiment?
Lines 145-146 – is there a possibility that moving the night-time seed plates away from the day-time plots may have influenced removal rates by, for example, moving the seed plates closer to/away from seed predators e.g. ant’s nests etc.?

Validity of the findings

The findings are interesting and well-reported, and any speculation is clearly labelled as such. The conclusions are relevant to the original research question and supported by the results. I only have two comments regarding the results, reported below. My only real issue is regarding Supplementary File 2. It is not clear why there is a set of NAs in the 8h-10h period in the Darmstadt data, as this is not mentioned in the manuscript. I understand that, while doing fieldwork, things outside our control occur which limit data collection, but a short explanation of why this occurred, or at least a statement that it occurred, will increase transparency. Otherwise, I believe the results are robust and fully support the conclusions drawn by the authors.

Further comments:
Lines 184-185 – how did removal rates change in time with habitat and seed type? A more detailed description of the results may be helpful.
Lines 254-256 – is there a way to access temperature data for the experimental period? It may provide another interesting layer of analysis.

Additional comments

This was an enjoyable study to read. I look forward to its publication.

Reviewer 2 ·

Basic reporting

Wehner et al.'s manuscript "Seed type, habitat and daytime influence post-dispersal seed removal in temperate ecosystems” attempts to disentangle the effects of seed traits, habitat type, and time of day on rates of seed removal for four species native to their study sites: Chelidonium majus, Lotus corniculatus, Tragopogon pratensis and Helianthus annuus. The authors attempt a tall order: to disentangle the effects of seed traits, habitat type, and time of day by presenting the results of two separate field studies. The first experiment compares seed removal across all 4 species in forests versus grasslands over a time period of 48 hours. The second experiment compares seed removal during the day versus night for the same suite of species in forests versus grasslands.

Overall, while the authors conclude that their study shows evidence that seed type and to a lesser extent habitat type influence seed removal, I think that their study design does not necessarily warrant such a general conclusion, and would benefit by adding in language to provide caveats to their overall conclusions. While I find their overall general conclusions problematic, I do think that they have completed two well-designed studies that fit well within this manuscript and within the scope of PeerJ.

Below, I list changes that I think should be made to improve the clarity of this manuscript and limit the scope to fit what is being tested with this study.

Please change “daytime” to “time of day” throughout the manuscript, including the title. The term daytime specifically refers to daylight hours, however, I think the authors really mean the time of day within a 24-hour period from the rest of the details provided.

Intro and background:
Consider starting the first paragraph of the intro at line 39 at post-dispersal seed removal. The focus of this manuscript is post-dispersal seed removal so it seems unnecessary to include what the authors mention before that for this manuscript.

Since the study focuses on seed traits, habitat type, and time of day I suggest restructuring the introduction around these factors. At the moment, the authors go into much detail about particular seed traits, but then include a single paragraph for both habitat type and time of day together. More detail is needed to build up to why the reader would predict a difference in seed removal rates as a result of these factors.

lines 84-86: The authors discuss how seed predators differ in preference for seed size. Seed dispersers also vary in in this preference. Consider adding in a sentence to describe this. Add additional references to back up your predictions (at the moment only Brown et al. 1975 is used here).

line 92: Remove “seeds” after T. pratensis.

lines 94-96: The authors mention that they compare seeds of C. majus and L. corniculatus to determine if there is an effect of elaiosomes. Given that so many other factors (chemistry, seed coat thickness, etc) potentially vary across these two species this is not a rigorous test of the effect of elaiosomes. Remove this line. I think that it is okay to speculate about this within your discussion with a thorough discussion of the potentially confounding caveats, but this study is not set up to test this question.

line 98: The word “greater” should be changed to larger.

Predictions about seed traits in general: I find this not to be a particularly good study design to be able to disentangle individual seed traits given the potential for confounding effects. Instead, the authors should make predictions about whether they expect difference overall among species and then describe why based on multiple traits at once.

Overall, the authors appear to conform to PeerJ standards and discipline norms when it comes to the basic structure of their manuscript. However, one change to the structure that may provide more clarity is in how the methods and results are structured around the two experiments. Use sub-titles to refer to each of the studies, instead of mentioning it in the text itself. Mirror this structure in the results.

I found that the figure captions need to be edited for clarity. For all figures, be clear about which experiment the data are from. Additionally, be clear about what you are showing with error bars (i.e. are they representing standard error, standard deviation, etc.?). What do the different parts of your boxplots indicate?

In figure 1, the authors refer to the colors as “dark grey” and “light grey”, however the color scheme appears to be white and grey.

Figure 1 shows seed removal as the proportion of seeds removed, while the next figures show the number of seeds removed. Be consistent in how you present this. Did you find the same patterns for proportion of seeds removed and total number of seeds removed in your statistical analyses?

It is unclear why time is presented on a logrhythmic scale on the x-axis of figure 2.

I find it difficult to distinguish among the points and lines in figure 2. This figure might benefit from use of color (making sure to use a color scheme that is color-blind friendly) to distinguish among seed types.

It would improve interpretation of patterns in the figures if there was some kind of indicator of the seed traits that were previously mentioned in the manuscript (seed size, dispersal syndrome, etc.). This could be done by reminding the reader of these in the figure captions or including it with pictures of the seeds or added words in the figure.

Dates in the raw data do not all appear to fall into June of 2018. Experiment 2 appears to have been completed in July of 2018. Confirm that the dates are correct in dataset from Experiment 2 and then make the necessary change to the methods section in the text.

It does appear that all of the data are present in the two supplements that are mentioned in the methods. However, in the discussion the authors mention observations of ants and slugs removing seeds yet do not provide data or description in the methods of when they looked at this.

Experimental design

Wehner et al., present primary research that appears to be well within the scope of PeerJ. Additionally, from what is presented by Wehner et al., it appears that their work is performed to a high technical and ethical standard.

Below I give provide a list of things that I think need clarification or need to be added to the methods description:

line 110-111: It is unclear why this study was done in three locations from what is written here. Did the authors choose the three locations because each had grassland and forest habitat? For replication?

It would be good to add more general details about your study locations. How far away from each other are these sites? Are they predominantly grasslands or forests? What is known about their history of land-use? Add more detail about how they are currently managed.

lines 111-112: What time of year relative to the growing season are the meadows mowed?

line 113: Be more specific here about what a moderate amount of forestry entails. What forestry management techniques were used? Are they the same across sites?

lines 117-120: Be consistent with the information you present for each species so that the species are more easily compared. For example, you state that the seeds of speces a and b are small, but then make no similar mention to size for species c and d. Additionally, you mention that species a is dispersed by arthropods and species c is wind-dispersed, but then do not mention dispersal mode for the rest of the species.

line 121-123: Change the wording here. Instead of stating that the dry mass was “averaged with standard deviation”, state that you calculated the mean and standard deviations of each species seed dry mass.

line 128: How many wells are in the plastic plate? It is unclear whether each seed is placed in an individual well. Overall, I’m having a hard time picturing what these looked like making it difficult to envision how I would replicate this setup. Could you add some detail about the brand of plastic plate that you used?

line 130: How did you make sure seed plates were placed flat on the ground? Additionally, it is unclear how you chose the location where you placed the plates. Were these locations random within a site?

line 154: It is unclear why the authors define removal rate as R = log(N removed/25+1). Authors need to provide clarification for why they use this calculation.

lines 159-165: Did the authors consider a non-linear fit for this analysis. When looking at figure 2, some of the species may be better fit by a different shape, but it doesn’t appear that the authors checked this.

line 161: Change “root-square” to square root. Add reason for transforming the data in this way.

line 163: It is unclear what the authors are specifically referring to when they say “individual effects”. Please clarify.

line 164 and 168. It is unclear how you are using one-way ANOVAs in these instances and why this was used. Please be more specific.

Note for statistical analysis in general: Authors need to be more specific about how they made sure their data fulfilled the assumptions of the statistical models that they used. They make no mention of this in their methods section. Additionally, while the authors mention interaction terms in their tables, they make no mention of including interactions in their methods.

Validity of the findings

line 173: It appears that the authors are using the term seed type instead of species. While it seems okay to speculate on the effect of variation among the different species as a potential result of the differences in their seeds, this study does not necessarily provide a rigorous test of this. There are lots of potential confounding variables that differ across the seeds of these four species. In order to provide rigorous evidence of the effects of seed type, the authors would need to add lots more species over a gradient of seed type variables. Change seed type to species.

Note for results in general: Authors need to describe the significant interaction terms present in table 2. Sometimes significant main effects cannot be safely interpreted when interaction terms are significant. Authors need to describe the significant interaction terms and provide a description of their significant main effects in the context of these significant interactions terms.

lines 209 and 215: The authors mention that they observed ants and slugs removing seeds, however nowhere in their methods or results do they mention that they observed who was removing seeds from their seed plates. For which study was this observed? Even if this was done opportunistically during the counting of seeds, it would benefit this manuscript to further describe this.

line 220: Change the term parachute to pappus. While these two terms are synonyms, pappus is more commonly used in the literature to describe this morphological feature on seeds that are wind dispersed.

lines 226-231: While seed size varies across your species, this trait is confounded by other traits such as morphology that lends seeds to being dispersed by animals (e.g. presence of an elaiosome) or wind (e.g. presence of pappus). Either remove this paragraph or add language that indicates this caveat.

line 232: Generally, it seems that you found not effect of habitat type, however the authors present this as having found little effect. The authors should change how this result is depicted here to be more in line with the evidence that they present which is that there was no effect of habitat type on seed removal for all but one species and that was only when the seed had been dehulled. Additionally, the discuss literature here about the effects of habitat type on seed removal, but it is unclear if they mean habitat type in general or if they are focusing on literature that measured seed removal in forests and grasslands. Add clarification.

---

## Round 0.2 · Minor Revisions

The authors and I really appreciate the revisions you've made to the manuscript. At this point, I'd still like for you to consider Reviewer 1s recommendation to reconsider your day / night analysis using light information as opposed to a 12 hour split between day and night.
Also, there are some relatively minor adjustments that should be made to the writing.

Reviewer 1 ·

Basic reporting

The language is clear and the literature has been widely covered. The article is easy to read and well-written. The addition of diagrams of seed type to the figures is very helpful. Some attention should be given to the spelling of author names in-text and in the reference list.

Specific comments:
Line 26 – “removed at significantly higher rates” instead of “significantly removed at higher rates”
Line 74, 433 – check spelling of “Xiao, Zhand/Zhang & Wang, 2005)
Lines 85-86 – it is not entirely clear how competition between species drives the temporal differentiation, perhaps add another sentence to explain this more.
Line 111 – “generally considered as trait associated” doesn’t make sense
Line 114 – explain why you expect seed removal to be higher at night
Line 119 – “interested in comparing the two” instead of “interested to compare”
Line 168 – “12 hours during the day” not “12 hours at day”
Line 170 – “12 hours during the night” not “12 hours at night”
Line 232 – “myrmecochorous because of the presence of” instead of “through the presence of”
Line 258 – add “for dispersal” to the end of “that depend on abiotic conditions”

Experimental design

The research question is clear and highlights the current gap in the knowledge relating to the predation and dispersal of seeds on the ground. The methods are coherent and clear, with the exception of the following:

Lines 168 – 170 – did day and night split evenly into 12 hours each? This is difficult to believe as the study locations and time would suggest that daylight hours would last longer than darkness (approximately 16 hours and 8 hours, respectively). Please instead report the correct periods of day/night -time and account for them in any statistical tests.

Validity of the findings

Study 1 is well-defined and the results are clear. However, I am hesitant to accept the findings from study 2, as the 12 hour day/12 hour night period does not appear to match the actual sunset/sunrise times in Germany during June and July. I suggest that the data is re-analysed using the actual daytime/night-time data, for currently the results do not correspond to the natural environment.

Reviewer 2 ·

Basic reporting

Lines 37-39 - Indicate that seed survival is also affected by seed size, dispersal mode, and annual seed production. As this sentence is currently written, it seems like the authors are drawing a distinction between the separate factors limiting seed dispersal and survival.

Line 18 - Consider removing “(micro-)” from before habitat. Given that the authors do not measure micro-habitat factors.

Line 53 - Change “consumption” to “predation” given that consumption itself doesn’t always mean seed death.

Lines 73-74 - Authors should elaborate further on how seed size affects seed dispersal (i.e. do smaller seeds move farther than larger seeds?). Authors could consider giving examples of when this has been found to be true.

Lines 75-82 - Consider offering examples of each instance. Why do the authors believe that habitat should have an effect.

Lines 83-86 - Further along in the Introduction, the authors make the prediction that seed removal will be highest at night (versus during the day). The authors should add more details to this paragraph to build up to that prediction.

Lines 96-97 - Elaborate here on the range of seed dispersal syndromes and range in size. This could potentially be added within parentheses after each term.


Line 133 - Change “dispersion” to “dispersed” to match grammar used in a, c, and d.

Lines 200-206 - Authors state that seed removal did not differ between habitats in lines 200-201, but then go on to say that intact H. annuus were removed faster in grasslands than in forests. Change the first sentence to indicate this result, otherwise it seems as if there was never a difference detected across habitat types.

Lines 263-264 - This sentence is a bit awkward since the authors start off by saying habitat has no effect and then follow that up with saying but it they found higher removal in grasslands. The sentence could be changed to something like “Typically, habitat had no effect on seed removal, although removal of intact H. annuus was higher in grasslands.

Line 277 - Consider removing the word “different” from before seed types. It will improve the flow of the sentence.

Comments on figures:
Figure 2 - Consider moving the legend such that it no longer overlaps data points. One option could be to move it to where the seed drawings and descriptions are now located.

Experimental design

Line 131 - Please report the source of your seeds (i.e. did the authors collect them, purchase them, etc?) to aid with providing sufficient detail for replication.

Line 146 - 161 - Here the authors should consider switching the order of these two paragraphs. As a reader, I was left wondering what a subplot was until the following paragraph, but starting with the description of the sites and layout of the plots would let the reader know right away.

Lines 168-171 - It is unclear if the authors used the same methodology for their seed plates in Experiment 1 to set out the seeds in Experiment 2. Please clarify. Additionally, the authors shift from using the term plate to using the term panel (line 170). Be consistent in terminology to improve clarity. How did the authors choose the location for seed plates in Study 2?

Lines 174-196 - Make sure you include your models’ interaction terms in your descriptions of your statistical model. Additionally, make sure your results include mentioning what your models are showing for the interaction terms. If interaction terms are explaining a significant amount of the variation in your data, you additionally need to explain whether you can interpret the effects significant main effects when significant interaction terms are present.

Lines 183-184 - I think the blocking factor (e.g. the random factor) in the model should instead be plot and not subplot. The authors should fix this or add to this sentence to justify why it should be subplot instead. Additionally, this description should match up with the data provided in the supplemental.

Validity of the findings

No additional comments here.

Additional comments

Did the authors consider the possibility of a non-linear best-fit line for the analysis of seed removal over time (those shown in figure 2). Add whether this was done to your methods/results.

The authors should still consider adding language somewhere that indicates that variation in the traits that they measured may not necessarily be causal owing to the potential collinearity with other unmeasured traits. Additionally, authors should add to the discussion to explain the possibility of whether they think the patterns would change if they added more species (i.e. would they expect to see the same general patterns if conducting this study at or closer to the scale of the whole plant community (i.e. what would be expected if they added more plant species?).

Authors should add transitions between paragraphs to improve flow of ideas from one paragraph to the next. Occasionally, there are places in the manuscript where grammar could be improved (e.g. lines 118-119 “Since we were interested to compare the two common habitats forest and grasslands…” could be changed to “Since we were interested in comparing the two common habitats forest and grasslands…”).

---

## Round 0.3 · accepted · Accept

Thanks for your careful revisions. I'm happy to accept your manuscript for publication in PeerJ at this point!